# Characterization and Experimental Use of Multiple Myeloma Bone Marrow Endothelial Cells and Progenitors

**DOI:** 10.3390/ijms252212047

**Published:** 2024-11-09

**Authors:** Filip Garbicz, Marcin Kaszkowiak, Julia Dudkiewicz-Garbicz, David M. Dorfman, Julia Ostrowska, Joanna Barankiewicz, Aleksander Salomon-Perzyński, Ewa Lech-Marańda, Tuyet Nguyen, Przemyslaw Juszczyński, Ruben D. Carrasco, Irena Misiewicz-Krzeminska

**Affiliations:** 1Harvard Medical School, 25 Shattuck Street, Boston, MA 02115, USA; filipa_garbicz@dfci.harvard.edu (F.G.);; 2Department of Pathology, Dana-Farber Cancer Institute, 450 Brookline Ave, Boston, MA 02115, USA; 3Department of Experimental Hematology, Institute of Hematology and Transfusion Medicine, Gandhi 14, 02-776 Warsaw, Poland; mgkaszkowiak@gmail.com (M.K.);; 4Department of Immunology, Medical University of Warsaw, Nielubowicza 5, 02-097 Warsaw, Poland; 5Broad Institute of MIT and Harvard, 415 Main St, Cambridge, MA 02142, USA; 6Department of Methodology, Medical University of Warsaw, Żwirki i Wigury 81, 02-091 Warsaw, Poland; julia.m.dudkiewicz@gmail.com; 7Department of Pathology, Brigham and Women’s Hospital, 75 Francis St, Boston, MA 02115, USA; 8Department of Hematology, Institute of Hematology and Transfusion Medicine, Gandhi 14, 02-776 Warsaw, Poland

**Keywords:** myeloma, plasma cell, endothelial, progenitors, cancer, microenvironment, bone marrow, scRNAseq, angiogenesis

## Abstract

Multiple myeloma (MM) is a plasma cell malignancy that resides within the bone marrow microenvironment, relying heavily on interactions with its cellular components. Among these, endothelial cells (ECs) play a pivotal role in MM progression and the development of therapeutic resistance. In this study, we analyzed publicly available single-cell RNA sequencing data to identify unique pathway activations distinguishing ECs from MM patients and healthy donors. We developed a novel protocol to isolate and culture endothelial progenitor cells (EPCs) and ECs directly from MM patient bone marrow, demonstrating their ability to promote myeloma cell proliferation. Validation studies confirmed that these MM-derived ECs exhibit angiogenic potential as well as the expression of characteristic endothelial lineage markers. These findings underscore the critical role of bone marrow ECs in the MM tumor microenvironment and highlight potential new therapeutic targets to disrupt MM progression.

## 1. Introduction

Multiple myeloma (MM) is a plasma cell neoplasm that develops within the bone marrow, where malignant plasma cells rely heavily on interactions with the surrounding microenvironment [1]. This niche comprises various cell types, including endothelial cells (ECs) [2,3,4], mesenchymal stem cells [5], and immune cells [6], which collectively support MM cell survival, proliferation, and drug resistance [1,2]. Previous studies have demonstrated that the genetic ablation of bone marrow endothelial progenitor cells (EPCs) can abrogate MM development [3], underscoring the importance of ECs in disease progression. Furthermore, increased bone marrow microvascular density correlates with MM progression from monoclonal gammopathy of undetermined significance (MGUS) and smoldering multiple myeloma (SMM) to active MM, and serves as a prognostic marker associated with poor outcomes [7].

Our prior work has shown that bone marrow ECs provide homing cues to MM cells, and that targeting the endothelial BCL9/eCypA axis can disrupt the supportive interactions between ECs and MM cells [2]. While recent single-cell RNA sequencing (scRNAseq) studies have advanced our understanding of the MM tumor microenvironment, they have largely focused on mesenchymal stem cells, leaving the EC fraction underexplored [5].

In this study, we leveraged a high-quality scRNAseq dataset of MM and normal bone marrow samples [5] to perform the first unbiased analysis of transcriptomic changes in MM-derived ECs compared to normal bone marrow ECs. Additionally, we developed a detailed protocol for the isolation and characterization of ECs from MM patients, proposing these cells as a novel in vitro model to investigate EC biology in the context of MM.

## 2. Results

### 2.1. Single-Cell RNAseq-Based Characterization of MM and HD ECs

To investigate the characteristics of endothelial cell populations in MM patients and healthy controls, we utilized four scRNAseq datasets from de Jong et al. [5], merging scRNA-seq datasets from CD45− cells, CD45+ CD38+ cells (non-PC), CD45+ CD38− cells, and CD45+ CD38+ MM plasma cells (Figure 1A; cluster annotations were preserved from the original study (Appendix A)). UMAP projections (Figure 1A) revealed the distinct clustering of cells across multiple datasets. ECs were identified via the specific expression of *CDH5* (green arrow), yielding 502 endothelial cells from healthy donors (HD ECs) and 1275 endothelial cells from MM patients (MM ECs), with distinct clustering suggesting transcriptional differences in disease vs. normal conditions. A cluster-specific gene expression dot plot recapitulated the original cluster assignment performed by the authors of the original datasets (Figure 1B). The EC cluster exhibited a high expression of known EC-associated markers such as *CDH5*, *PECAM1*, *FLT1*, *KDR*, and *ERG* (Figure 1B,C), while other immune and stromal cells expressed their characteristic lineage-defining genes. *PIM3*, an oncogenic kinase previously identified by us as a protein marker of MM ECs using immunohistochemistry [8,9], was highly expressed at the mRNA level across the majority of the cell clusters. While its average expression was highest in ECs compared to other stromal CD45− cells, it was also detected in immune cell clusters, particularly in T cells (Figure 1B). Further analysis (Figure 1C) revealed robust EC marker expression in both the MM patients and the HDs and the absence of hematopoietic markers like *PTPRC* (the gene encoding CD45), ensuring that the analysis was not confounded by the presence of hematopoietic cells expressing endothelial markers [10]. No endothelial progenitor cell clusters were found, so subsequent bioinformatic analysis focused on the mature ECs.

A pseudobulk differential gene expression analysis revealed significant transcriptional differences between the ECs from the MM patients and the HDs (Figure 2A). A gene set enrichment analysis (GSEA) identified key pathways enriched in MM ECs compared to HD ECs, such as epithelial–mesenchymal plasticity (EMP), TGF-beta signaling, and angiogenesis (Figure 2B). An analysis of transcription factor targets revealed the enrichment of AP1- and BACH1/2-dependent genes, indicative of angiogenic stimulation [11,12] (Figure 2C). Additional GSEA on the GO and Reactome collections revealed MM EC-specific enrichment in the adherent junction, integrin signaling, and mesenchymal cell activity, while the genes involved in cytoplasmic translation were expressed at higher levels in the HD ECs (Figure 2D,E). A complementary Metascape-based clustering of ontology terms [13] showed that the MM ECs exhibited a positive enrichment in co-clustering terms, such as blood vessel development, VEGFA/VEGFR2 signaling, kinase activity, the regulation of cytoskeleton organization, and migration and signaling by Rho GTPases (Figure 2F). In contrast, the HD ECs exhibited enrichment in eukaryotic translation termination and scavenger receptor activity (Figure 2G). The visualization of the leading-edge genes in Hallmark gene sets positively enriched in the MM ECs identified several potential drivers of mesenchymal plasticity, such as *VIM* [14], *CTNNB1* [15], *SMAD3* [16], *MYC* [17], and *IL6* [18] (Figure 2H). *VIM*, along with other candidate EMP drivers such as *STC1* [19] and *SERPINE1* [20], was also among the top upregulated genes in the pseudobulk differential gene expression analysis (Figure 2A). On the other hand, MM ECs showed a loss of expression of several regulators of endothelial cell identity and activity, such as *HES1* [21], *PTCH2* [22], *PIK3IP1* [23], and *BTG2* [24] (Figure 2A).

### 2.2. Single-Cell RNAseq-Based Characterization of Cell–Cell Interactions in MM and HD ECs

Next, we examined the cell–cell communication dynamics of ECs in MM bone marrow and compared them to those of HDs. Using ligand–receptor interaction analysis, we inferred both efferent (outgoing) and afferent (incoming) signaling between ECs and other cell types within the bone marrow microenvironment (Appendix A and Figure 3).

The number of predicted cell–cell interactions was significantly higher across many clusters in the MM bone marrow compared to the HD bone marrow (Appendix A). Notably, mesenchymal stem cell (MSC) clusters 1–5 and endothelial cells exhibited the largest increase in predicted interactions in the MM bone marrow (Appendix A) relative to the HD (Appendix A). The analysis of aggregated signaling pathways (Appendix A) revealed major differences between the MM ECs and HD ECs, including pathways predicted to be active exclusively in MM bone marrow. Robust interactions were predicted between the MM endothelial ligands such as PDGFA, LAMA4, JAM3, COL4A1, CD46, and ANGPTL4, and their corresponding receptors on other bone marrow cell types (Figure 3A). For instance, PDGFA was predicted to primarily affect MSCs, which expressed high levels of PDGFRA/PDGFRB [5], while COL4A1 was predicted to engage the SDC1 (CD138) receptor on malignant plasma cells. Other efferent pathways included ANXA1/FPR1 and CD99, predicted to influence receptors on immune cells such as monocytes.

In addition, signaling directed toward MM ECs showed a marked increase, with notable enrichment in the angiogenesis-related pathways. VEGF signaling, mediated by ligands such as VEGFA/B/C and PGF, was predicted to activate receptors VEGFR1/2/3. Inflammatory signaling via JAG1-NOTCH, TNF, and SPP1-CD44 was also enhanced in MM ECs compared to HD ECs. Interestingly, the majority of the proangiogenic signals targeting ECs in MM were predicted to originate from MSCs, not malignant plasma cells. While plasma cells were predicted to promote endothelial growth through VEGFB secretion, the primary sources of VEGFA, VEGFC, PGF, and angiopoietin-like cytokines (ANGPTL2/4) were MSCs, particularly the MM-exclusive MSC1 cluster.

### 2.3. Isolation and Ex Vivo Characterization of ECs from MM Patients

We developed and optimized a protocol to reliably isolate and characterize both early outgrowth endothelial cells (composed primarily of endothelial progenitor cells) [25] as well as mature endothelial cells from bone marrow aspirates of MM patients (Figure 4A). The detailed protocol can be found in the Materials and Methods section. Briefly, after density gradient centrifugation to separate mononuclear cells, CD138+ plasma cells were positively selected and removed. The remaining CD138− cells were cultured on fibronectin-coated plates in endothelial growth media for 7–10 days, facilitating the attachment, colony formation, early cellular outgrowth, and subsequent differentiation and proliferation of mature ECs. Over the course of 9 days, CD138− cells cultured on fibronectin-coated plates progressively adopted an endothelial-like morphology, as shown in the bright-field images (Figure 4B). By day 9, the cells formed sparse colonies assuming the growth pattern of a confluent monolayer with a cobblestone-like appearance typical of endothelial cells, confirming successful culture and expansion. In some MM patients, bone marrow cells also formed mesenchymal-like colonies with spindle-shaped cells, often growing in multiple layers, emphasizing the necessity of CD31-positive selection to ensure purity. CD31+ ECs were purified using CD31-specific magnetic beads. To evaluate the angiogenic potential of the isolated cells, tube formation assays were performed in Matrigel. The bone marrow endothelial cell line (BMEC60), along with two independent MM endothelial cell populations (MMEC1 and MMEC2), demonstrated robust tube formation and tip cell-like structures, characteristic of active angiogenesis (Figure 4C). By contrast, the bone marrow stromal cells (BMSC1 and BMSC2) showed much less pronounced tube formation, supporting the endothelial identity of the MMECs. Flow cytometry analysis confirmed the endothelial identity of the isolated MMECs. Cells were gated for live single cells, and histograms showed the strong expression of endothelial markers, including CD31, VEGFR1, and CD144 (encoded by *CDH5*), compared to IgG controls (Figure 2D). This validated the purity and identity of the isolated ECs.

We then asked the question whether these cultured MM ECs resemble in situ bone marrow endothelial cells from MM patients. To address this, we measured the expression levels of a gene panel using real-time quantitative PCR (Figure 5A). RNA was isolated from the early outgrowth cells (EOCs) that appeared in the fibronectin-coated wells after 5 days of culture, as well as the magnetically sorted mature endothelial cells (CD31+) and the remaining mesenchymal cells (CD31−). Two endothelial cell lines (BMEC60, HUVEC) were used as positive controls. We found that EOCs expressed high levels of *PTPRC* (encoding CD45), while other cell types exhibited near undetectable levels of this transcript (Figure 5A), suggesting that EOCs may contain early endothelial progenitor cells, which have been described to be CD45+ [26].

In addition, we observed the expressions of *CD34*, *PECAM1*, *CDH5*, *FLT1* and *KDR* in both the EOC and CD31+ populations, with a trend towards lower transcript levels in the CD31− fraction. *PROM1* (encoding CD133) was expressed at overall higher levels in EOCs than in CD31+ or CD31− cells (Mann–Whitney test, *p* < 0.0001), consistent with CD133’s role as a marker for endothelial progenitor cells [27,28,29]. Both the EOC and CD31+ populations expressed genes characteristic of type H ECs (*APLNR*, *EFNB2*, *SOX17*, *FLT1*), as well as type L ECs (*STAB2*, *VCAM1*) [30]. *KDR* (encoding VEGFR2) and *PDGFA* (Appendix A) were expressed in both of the EOCs and CD31+ cells, with clearly higher levels in the MM-derived CD31+ cells compared to the HDs. These findings confirm the predominantly endothelial phenotype of both EOC and CD31+ populations.

We then evaluated their potential to support the survival and proliferation of ex vivo cultured CD138+ MM cells (Figure 5B). While the cells cultured in an empty medium did not proliferate, the conditioned medium from the MM CD31+ cells, BMEC60 cells, and MM CD31− cells supported the growth of patient-derived MM cells (*p* < 0.001).

Given our scRNA-seq characterization of MM endothelial cells (ECs), which indicated the increased expression of epithelial–mesenchymal plasticity-related genes (*VIM*, *STC1*), we investigated whether the in vitro MM EOCs and CD31+ ECs recapitulated this process when compared to the HD bone marrow cells. While there was a trend towards higher *STC1* expression in MM-derived cells, compared to HD, the differences were not statistically significant. Moreover, *VIM* expression did not align with the scRNA-seq findings.

A gene set enrichment analysis for transcription factors (Figure 2C) suggested the enrichment of AP1 target genes in MM ECs compared to HD. Since the AP1 activity in ECs was shown to be inhibited by the BET inhibitor JQ1 [31], we tested whether JQ1 could reverse the mesenchymal features of in vitro cultured MM CD31+ cells. However, JQ1 treatment did not affect the morphology of MM CD31+ cells (Figure 6B). While JQ1 downregulated the expression of *FLT1* and *KDR*, known endothelial super-enhancer genes, dependent on BET-domain proteins [32,33], it did not affect the expression of either *STC1* or *VIM*.

## 3. Discussion

In this study, we conducted a comprehensive analysis of the bone marrow endothelial cells (ECs) derived from multiple myeloma (MM) patients and healthy donors (HDs) (Figure 1, Figure 2 and Figure 3) using previously published single-cell RNA sequencing (scRNA-seq) datasets [5]. Our findings revealed significant transcriptional differences between MM-derived ECs and normal bone marrow ECs, highlighting pathways associated with epithelial–mesenchymal plasticity (EMP), TGF-beta signaling, and angiogenesis, as well as actin cytoskeleton and Rho protein activity (Figure 2). These results point at a more activated, proangiogenic phenotype of MM ECs, consistent with their expansion at advanced disease stages [2,34]. The overexpression of mesenchymal genes, such as *VIM* or *STC1*, suggests that the MM ECs undergo EMP or endothelial–mesenchymal transition (EndMT), a process often associated with cancer progression [35]. Interestingly, increased protein levels of Vimentin have already been associated with MM ECs and proposed as an MM-associated pathogenic mechanism [36]. However, this is the first study to detect a broader transcriptional signature characterizing MM EC transitioning from epithelial to a more mesenchymal state. Leveraging our scRNAseq analysis results, we also predicted enhanced cell–cell interactions (Figure 3) mediated by ligands such as PDGFA, angiopoietin-like, and other angiogenic factors, underscoring the complex communication between ECs and other cells within the MM bone marrow microenvironment.

Our ligand–receptor interaction analysis demonstrated that the MM ECs exhibited increased outgoing and incoming signaling pathways compared to HD ECs (Figure 3, Appendix A). Specifically, the MM ECs were predicted to produce higher levels of PDGFA, LAMA4, JAM3, COL4A1, CD46, and ANGPTL4, which may have interacted with the receptors on the mesenchymal stem cells (MSCs), malignant plasma cells, and immune cells. For instance, PDGFA secreted by ECs was predicted to primarily affect MM MSCs expressing PDGFRA/PDGFRB, potentially enhancing stromal support for MM cells. This cytokine axis has not been explored experimentally before and will be verified in a follow-up study. Discovering ways to target MSCs in MM is of high importance since these cells have been shown to be highly supportive towards myeloma cells [37,38,39,40]. Similarly, COL4A1 interactions with SDC1 (CD138) on malignant plasma cells may facilitate cancer cell adhesion and survival. These findings warrant further functional validation in subsequent studies.

Incoming signals targeting MM ECs were also markedly increased. VEGF signaling mediated by ligands such as VEGFA/B/C and placental growth factor (PGF) was predicted to engage VEGF receptors on ECs, promoting angiogenesis. Inflammatory signaling via TNF, and SPP1-CD44 was also enhanced, suggesting that MM ECs are responsive to pro-angiogenic and pro-inflammatory cues within the microenvironment. Interestingly, one of the ligand–receptor pairs predicted to be activated in MM ECs compared to HD ECs is JAG-NOTCH. The Jagged–Notch pathway has previously been linked to EMP [41] as well as MM progression [42] and therefore warrants prioritization during the exploration of MM microenvironment-targeting strategies in future studies. Notably, MSCs, rather than malignant plasma cells, were the primary source of these pro-angiogenic signals, highlighting the crucial role of stromal cells in MM progression. Our analysis results could also explain the lack of response to bevacizumab [43], a VEGFA inhibitor, since other types of vascular growth factors are highly expressed in the MM tumor microenvironment, leading to possible compensation and drug resistance.

A central contribution of this study is the development of a new protocol for isolating and culturing ECs directly from MM patient bone marrow (Figure 4). This model enables the study of MM ECs in a controlled environment and has demonstrated the ability of these ECs to promote myeloma cell proliferation both in vitro (Figure 5B) and in vivo [2]. This optimized model facilitates the in vitro differentiation of endothelial progenitor cells (EPCs) into mature ECs (Figure 5A), establishing a valuable tool for studying endothelial biology in MM and assessing anti-angiogenic therapies.

Despite its strengths, this model has limitations. One key limitation is its inability to fully replicate the complex cytokine milieu and extracellular matrix of the MM bone marrow microenvironment, which may limit its effectiveness in studying EMP. Cytokines and growth factors play crucial roles in modulating EMP [44], and their absence may explain why certain gene expression changes, such as elevated *VIM* and *STC1* levels observed in scRNA-seq data, were not reproduced in vitro (Figure 6A). Our attempt to reverse EMP in MM ECs using JQ1 [45] was also unsuccessful (Figure 6B,C), despite JQ1 targeting BRD4, a known activator of EMP transcription factors in some cancers [46]. The downregulation of super-enhancer-regulated endothelial genes confirmed JQ1 activity in the primary ECs (Figure 6D), suggesting that alternative therapeutic strategies or more complex models may be required to accurately study this plasticity in MM ECs.

Nevertheless, our model remains highly valuable for specific applications. Its strong angiogenic activity, as shown in tube formation assays (Figure 4C), makes it an ideal platform for testing anti-angiogenic therapies targeting pathways such as VEGF and PDGF. These therapies could be evaluated for their potential to disrupt EC-mediated support for MM cells or other cells within the MM microenvironment, such as MSCs. Additionally, our protocol’s ability to differentiate early outgrowth cells (EOCs) into mature ECs provides a useful tool for exploring endothelial biology in MM.

To address the limitations of our current model, future research should explore the development of three-dimensional organoids that better mimic the MM bone marrow environment. Incorporating extracellular matrix components and cytokine gradients [47] could enhance the physiological relevance of this model, allowing for a more comprehensive study of EMP and cell–cell interactions. Murine models also remain indispensable for in the vivo studies of MM [48,49,50], enabling the investigation of complex systemic interactions and the evaluation of therapeutic interventions in the context of a whole organism.

In conclusion, our findings underscore the critical role of bone marrow ECs in supporting MM progression through enhanced cell–cell interactions and the activation of angiogenic pathways. While our current in vitro model has limitations, it provides a valuable tool for specific studies and sets the stage for developing more comprehensive models that can facilitate the discovery of effective therapies against MM. Future studies should focus on refining these models and further characterizing the proposed cell–cell interactions within the MM bone marrow niche to identify novel therapeutic targets.

## 4. Materials and Methods

### 4.1. Data Acquisition and Preprocessing

Single-cell RNA sequencing (scRNA-seq) datasets were obtained from de Jong et al. [5], encompassing bone marrow samples from multiple myeloma (MM) patients and healthy donors (HDs). Specifically, we utilized datasets comprising CD45− cells, CD45+ CD38+ cells (non-PC), CD45+ CD38− cells, and CD45+ CD38+ MM plasma cells. The datasets were provided as Seurat objects and were merged using Seurat version 5.1.0 [51].

### 4.2. Single-Cell RNA-Seq Data Processing and Analysis

Cells with low-quality metrics were filtered out based on the criteria defined in the original study [5]. The Seurat objects were normalized using the NormalizeData function with default parameters. Uniform Manifold Approximation and Projection (UMAP) was performed using RunUMAP for dimensionality reduction and visualization. Cluster identities were assigned based on canonical marker genes and the annotations provided in the original datasets. Endothelial cells (ECs) were identified by the expression of specific markers such as *CDH5*, *PECAM1*, and *FLT1*. The cells expressing these markers and lacking expression of hematopoietic marker *PTPRC* (CD45) were subsetted for further analysis. To identify the differentially expressed genes (DEGs) between MM-derived ECs (MM ECs) and HD-derived ECs (HD ECs), a pseudobulk approach was employed. Gene expression counts were aggregated using the AggregateExpression function, and grouping by “state” (MM or HD) and “source” (patient identifier). DESeq2 (version 1.46.0) with Benjamini–Hochberg multiple testing correction was conducted using the FindMarkers function. A full list of differentially expressed genes is available in Appendix A.

### 4.3. Gene Set Enrichment Analysis

Gene set enrichment analysis (GSEA) was conducted using the fgsea package (1.32.0) [52] and the clusterProfiler (4.15.0) [53]. Gene sets were obtained from the Molecular Signatures Database (MSigDB), including Hallmark, Reactome, Gene Ontology Biological Processes (GO), and transcription factor targets. Ranked gene lists were generated based on the stat parameter. Enrichment scores and normalized enrichment scores were calculated with 10,000 permutations, considering gene sets with a minimum size of 10 genes.

### 4.4. Cell–Cell Communication Analysis

Cell–cell interaction networks were inferred using the CellChat package 5 [54]. The built-in CellChatDB.human database was used as the reference for ligand–receptor interactions. Overexpressed genes and interactions were identified using identifyOverExpressedGenes and identifyOverExpressedInteractions. Communication probabilities were computed with computeCommunProb, and significant interactions were filtered with filterCommunication (minimum of 10 cells per group). A pathway level communication analysis was performed with computeCommunProbPathway, and aggregated networks were visualized using circle plots and heatmaps.

### 4.5. Isolation and Culture of Endothelial Cells from Bone Marrow Samples

Bone marrow aspirates were obtained from the newly diagnosed MM patients following informed consent and in accordance with the institutional ethical guidelines, as approved by the Institute of Hematology and Transfusion Medicine Bioethical Committee (43/2016) and DFCI IRB #07-150. Mononuclear cells were isolated using density gradient centrifugation with Histopaque-1077 (Sigma-Aldrich, Saint Louis, MO, USA) performed for 20 min at 250 RCF. No RBC lysis was performed on the samples. The CD138⁺ plasma cells were purified using magnetic-activated cell sorting (MACS) with CD138 MicroBeads (Miltenyi Biotec, Bergisch Gladbach, Germany). The remaining CD138^−^ fraction was cultured on fibronectin-coated 24-well plates (Sigma-Aldrich) in Endothelial Basal Medium-2 (EBM-2; Lonza, Portsmouth, NH, USA), supplemented with growth factors from the EGM-2 BulletKit (Lonza, Portsmouth, NH, USA) and 10% human AB serum (Fisher Scientific, Pittsburgh, PA, USA). Typically, one bone marrow sample was divided and seeded across 12 wells of a 24-well plate. Fibronectin (F2006-5MG, Sigma-Aldrich, Saint Louis, MO, USA) was applied at a concentration of 50 μg/mL, with 300 μL per well, and incubated in a cell culture incubator for 1 h. After incubation, the fibronectin solution was aspirated, and the plate was air-dried. At 24 h post-seeding, the medium was gently replaced, removing the suspended cells while adherent cells remained attached. Thereafter, the medium was exchanged every 48 h. Early outgrowth endothelial colonies were observed between 5 and 7 days, and around day 10, subconfluent cell sheets were visible. Based on our experience, the success rate for generating these colonies was approximately 90%. CD31+ endothelial cells were further purified using CD31 MicroBeads (Miltenyi Biotec, Bergisch Gladbach, Germany) and cultured under the same conditions (EBM-2 + EGM-2) on fibronectin-coated plasticware. Purity was confirmed via flow cytometry, with a high expression of endothelial markers CD31, VEGFR1, and CD144. The BMEC60 cell line was kindly provided by Ellen van der Schoot [55], and HUVEC-TERT2 cells were obtained from ATCC. Both endothelial cell lines were cultured on 0.2% gelatin-coated plates in EBM-2 (Lonza, Portsmouth, NH, USA) with EGM-2 BulletKit (Lonza, Portsmouth, NH, USA) supplementation. All the cells were maintained in a standard incubator at 37 °C with 5% CO₂ and 95% relative humidity. A conditioned medium was obtained by incubating adherent, confluent cells (pre-washed with PBS) in OptiMEM (Gibco, Grand Island, NY, USA) for 24 h.

### 4.6. Endothelial Cell Characterization

A morphological assessment was performed using bright-field microscopy (PrimoVert Inverted Phase Contrast Microscope; Zeiss, Oberkochen, Germany). Tube formation assays were conducted by seeding ECs (40,000 cells) onto Matrigel (Corning, Corning, NY, USA) solidified in 24-well plates and incubating it in a 37 °C cell incubator for 6 h. Tube-like structures were visualized and imaged using phase-contrast microscopy (PrimoVert Inverted Phase Contrast Microscope; Zeiss, Oberkochen, Germany). Flow cytometry analyses were carried out on a BD FACSCanto II cytometer (Becton Dickinson, Franklin Lakes, NJ, USA). The cells were stained with fluorophore-conjugated antibodies against CD31 (WM59), VEGFR1 (A16083C), CD144 (BV9), and corresponding isotype controls (BioLegend, San Diego, CA, USA). The data were analyzed using FlowJo version 10 (FlowJo LLC, Ashland, OR, USA).

### 4.7. Gene Expression Validation

Quantitative real-time PCR (qRT-PCR) was used to validate the expression of key genes identified from scRNA-seq analysis. Total RNA was extracted using Trizol and cDNA was synthesized using the SuperScript IV First-Strand Synthesis System (Thermo Fisher Scientific, Waltham, MA, USA). qRT-PCR was performed with a SYBR Green Master Mix (Bio-Rad, Hercules, CA, USA) using a CFX96 Touch Real-Time PCR Detection System (Bio-Rad, Hercules, CA, USA ). The gene expression levels were normalized to the expression of 3 housekeeping genes *YWHAZ*, *ACTB* and *RNA18S* and calculated using the 2^(−ΔCt)^ method. Oligonucleotide sequences have been listed in Appendix A.

### 4.8. Statistical Analysis

All the statistical analyses utilizing scRNAseq data were performed using R version 4.4.1 and associated packages. Differential expression results were corrected for multiple testing using the Benjamini–Hochberg method. The statistical significance was set at an adjusted *p*-value < 0.05. The qPCR results’ significance was assessed using Brown–Forsythe and Welch ANOVA tests with Dunnett’s T3 multiple comparisons test, a Mann–Whitney test, or an unpaired t-test in GraphPad Prism 9.5.1 (GraphPad Software, Boston, MA, USA).

## Figures and Tables

**Figure 1 ijms-25-12047-f001:**
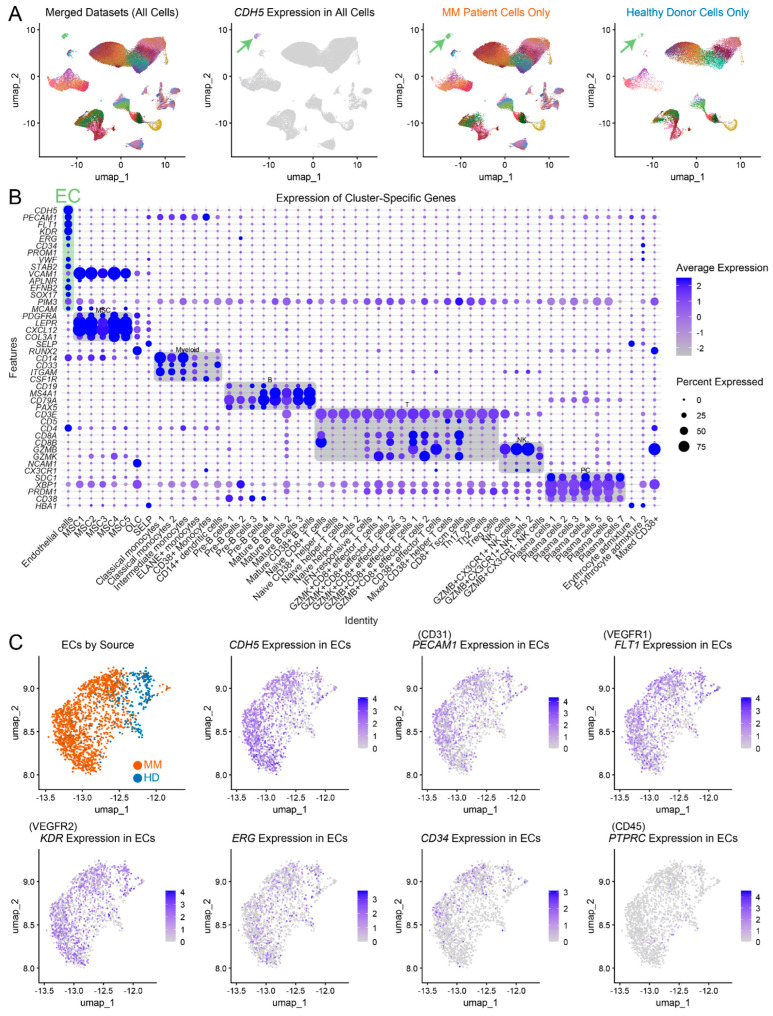
scRNA-seq analysis of ECs from MM patients and HDs. (**A**) UMAP plots showing merged datasets (**left**), *CDH5* expression (**middle**), and separated MM patient and healthy donor cells (**right**), highlighting endothelial cell clusters (green arrows). (**B**) Dot plot of cluster-specific gene expression, showing endothelial cell marker expression (*CDH5*, *PECAM1*, *FLT1*, *KDR*, *ERG*, *CD34*, *VWF*, *STAB2*, *APLNR*, *EFNB2*, *SOX17*, *PIM3*) in EC cluster (green). (**C**) UMAP projections of ECs from MM patients (orange) and controls (blue), with expression of key cellular markers.

**Figure 2 ijms-25-12047-f002:**
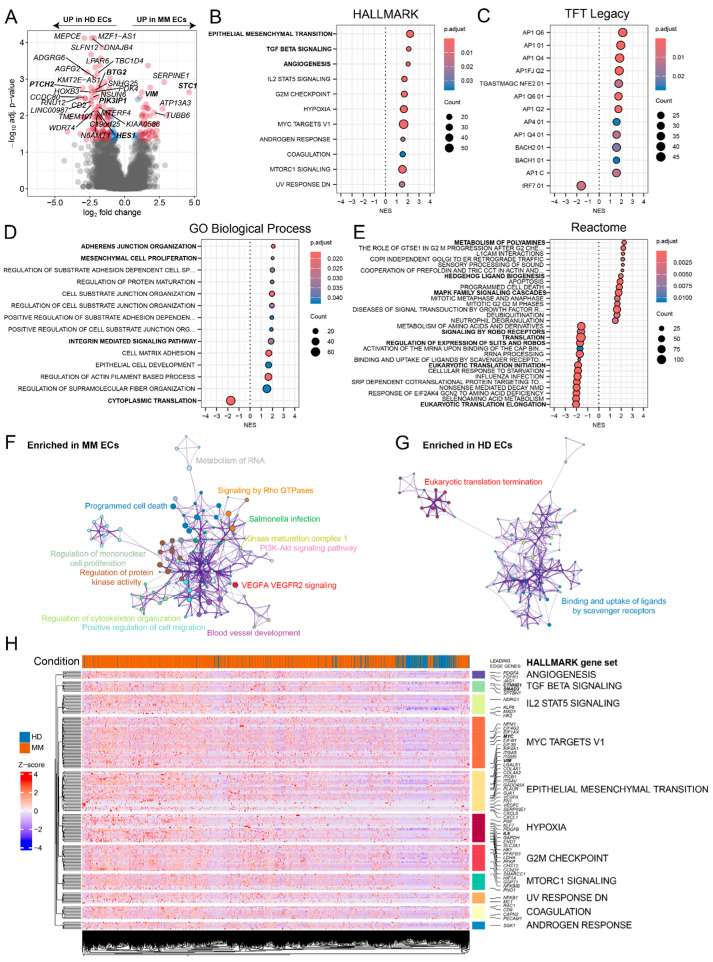
Differential expression and gene set enrichment analysis of MM vs. HD ECs. (**A**) Volcano plot highlighting significantly upregulated or downregulated genes in MM vs. HD ECs (red labels). (**B**–**E**) Gene set enrichment analysis (GSEA) results from Hallmark, TFT LEGACY, GO Biological Process, and Reactome collections, where dot size represents gene count in each pathway, and color indicates adjusted *p*-value. NES is on x-axis of each dot plot. (**F**,**G**) Metascape network of top positively enriched pathways in MM ECs (**F**) and HD ECs (**G**). (**H**) Heatmap of leading-edge genes from enriched Hallmark gene sets across control (blue) and myeloma (orange) ECs, with Z-score normalized expression (red = upregulation, blue = downregulation).

**Figure 3 ijms-25-12047-f003:**
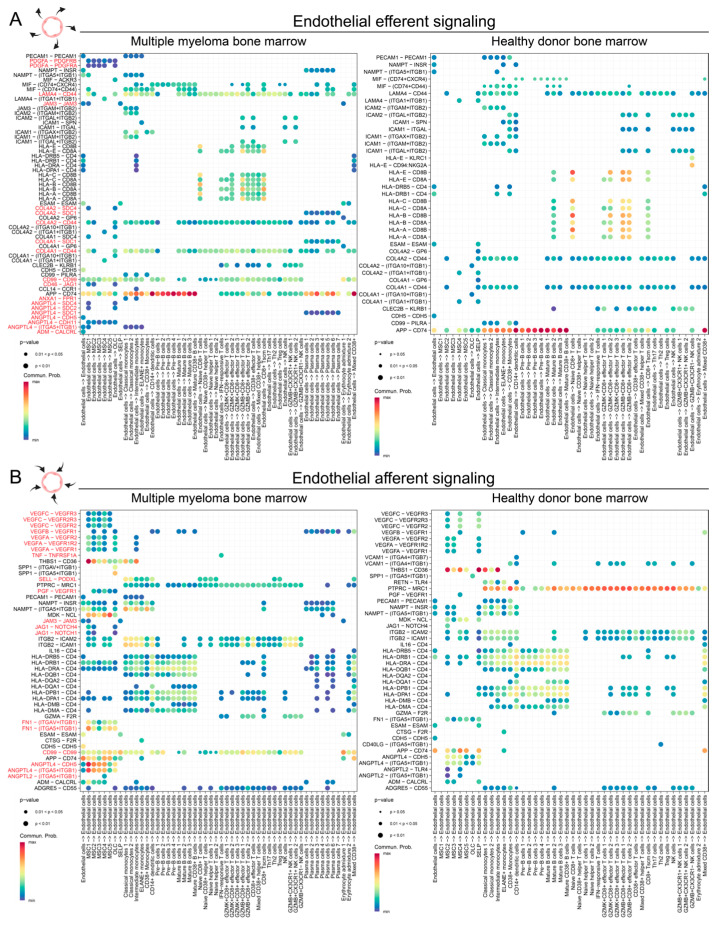
Cell–cell interaction analysis of endothelial efferent and afferent signaling in MM and HD bone marrow ECs. (**A**) Endothelial efferent signaling. Dot plots depicting predicted efferent signaling from endothelial cells to other cell types in bone marrow of MM patients (**left**) and HDs (**right**). (**B**) Endothelial afferent signaling. Dot plot showing predicted afferent signaling to endothelial cells from other cell types in bone marrow of MM patients (**left**) and HDs (**right**). In all cases, dot size represents interaction *p*-value, and color indicates communication probability, with higher values shown in warmer colors. Red labels highlight pathways enhanced in MM ECs.

**Figure 4 ijms-25-12047-f004:**
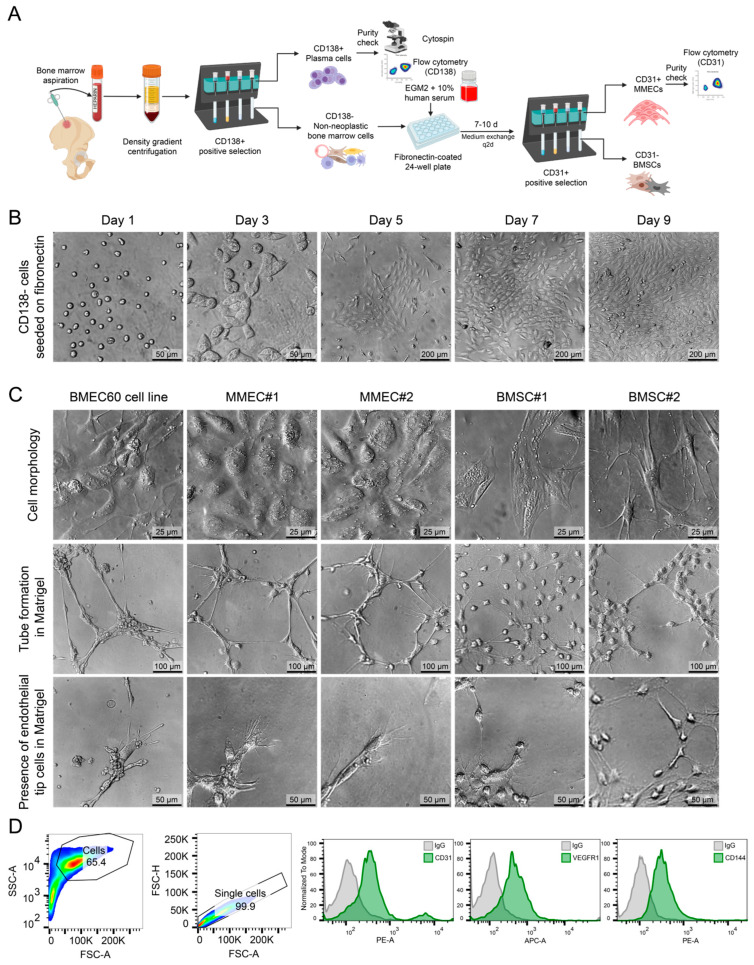
Isolation and characterization of EPCs and ECs from MM patient bone marrow biopsies. (**A**) Workflow illustrating isolation of CD31+ ECs and CD138+ PCs from MM patient bone marrow aspirates. Bone marrow mononuclear cells were separated via density gradient centrifugation, followed by CD138+ positive selection. Non-PC bone marrow cells were then cultured on fibronectin-coated plates for 7–10 days, with subsequent CD31+ endothelial cell positive selection using magnetic beads. (**B**) Bright-field images showing morphological changes in CD138− bone marrow cells on fibronectin-coated plates over 9 days, forming an endothelial-like monolayer by day 9. (**C**) Bright-field images of bone marrow endothelial cells (BMEC60), multiple myeloma endothelial cells (MMEC1, MMEC2), and bone marrow stromal cells (BMSC1, BMSC2) in culture, with tube formation and tip cell-like structures shown in Matrigel assays. (**D**) Flow cytometry plots showing cell gating and histograms displaying expression of CD31, VEGFR1, and CD146 compared to IgG controls, confirming endothelial identity. EGM2—endothelial growth medium 2. Parts of figure created in Biorender.

**Figure 5 ijms-25-12047-f005:**
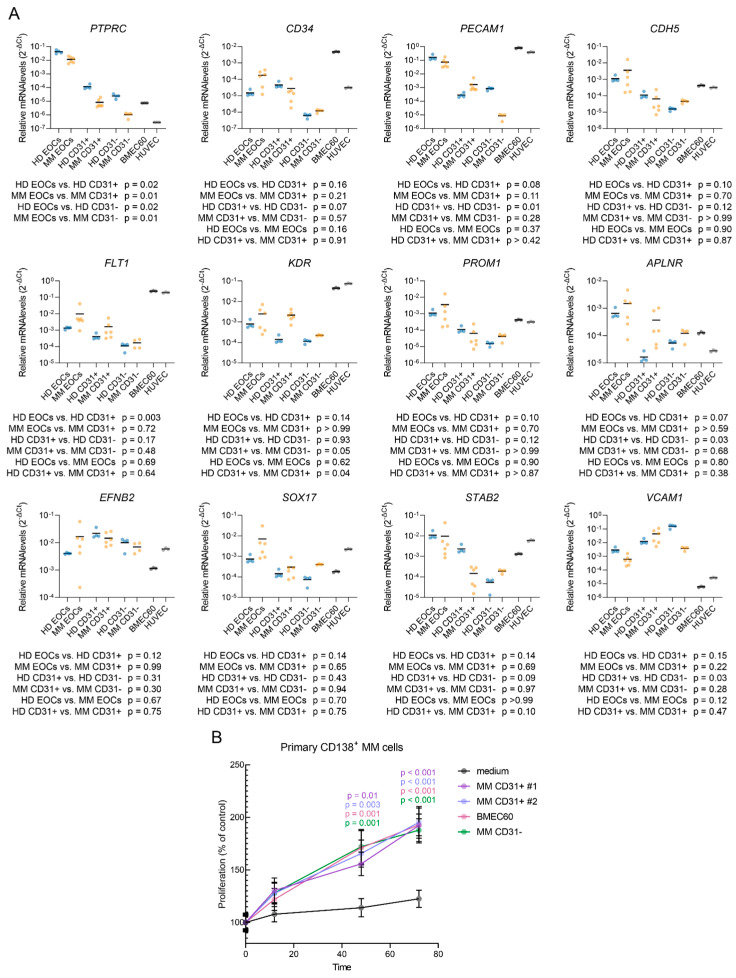
Expression of endothelial and mesenchymal markers in different cell populations and the effect on primary CD138+ MM cell proliferation. (**A**) qPCR analysis of the cell lineage markers (*PTPRC*, *CD34*, *PECAM1*, *CDH5*, *FLT1*, *KDR*, *PROM1*, *EFNB2*, *STAB2*, *SOX17*, *APLNR*, and *VCAM1*) across different cell populations, including HD-EOCs, MM-EOCs, HD-CD31+, MM-CD31+, HD-CD31−, MM-CD31− and endothelial cell lines. Statistical significance was assessed using Brown–Forsythe and Welch ANOVA tests with Dunnett’s T3 correction for multiple comparisons. HD EOCs *n* = 4; MM EOCs *n* = 6; HD CD31+ *n* = 4; MM CD31+ *n* = 6; HD and MM CD31− *n* = 4; cell lines *n* = 3. Horizontal line indicates the mean. Blue dots indicate HD, yellow dots indicate MM, and grey dots indicate cell lines. (**B**) Proliferation of the primary CD138+ MM cells in the presence of various cell types. MM CD31+ cells (#1 and #2), MM CD31+ cells, and BMEC60 cells were used to collect conditioned medium, which was then added to the primary CD138+ MM cells. Proliferation was measured over time (12 h, 48 h, 72 h). Data are presented as a percentage of control proliferation. Error bars represent standard deviation.

**Figure 6 ijms-25-12047-f006:**
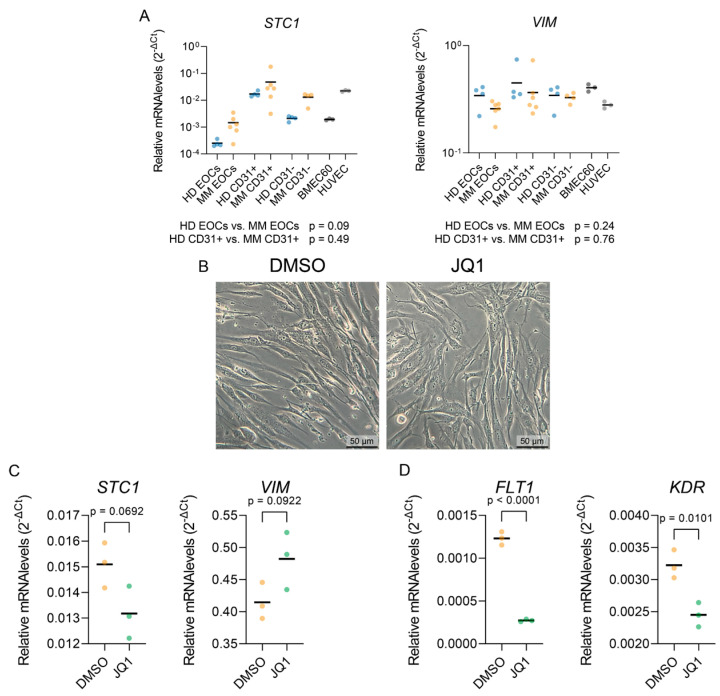
JQ1 treatment’s effect on the expression of select mesenchymal and angiogenic markers. (**A**) qPCR analysis of *STC* and *VIM* expression levels in indicated cell populations. Statistical significance was assessed using Brown–Forsythe and Welch ANOVA tests with Dunnett’s T3 correction for multiple comparisons. HD EOCs *n* = 4; MM EOCs *n* = 6; HD CD31+ *n* = 4; MM CD31+ *n* = 6; HD and MM CD31− *n* = 4; cell lines *n* = 3. Horizontal line indicates mean. Blue dots indicate HD, yellow dots indicate MM, and grey dots indicate cell lines. (**B**) Representative bright-field images of cells treated with DMSO (**left**) or JQ1 500 nM (**right**) for 24 h (**C**) qPCR analysis of expression levels of mesenchymal markers *STC1* and *VIM* in cells treated with DMSO or JQ1 (500 nM) for 24 h. (**D**) qPCR analysis of MM-upregulated angiogenic markers *FLT1* and *KDR* upon DMSO or JQ1 (500 nM) treatment for 24 h. Unpaired *t*-test has been used to assess statistical significance. *n* = 3. Horizontal bar represents mean.

## Data Availability

The original datasets are available at https://www.bmbrowser.org/de-jong-2021, accessed on 1 August 2024.

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
