# Peer review of "Characterization and Experimental Use of Multiple Myeloma Bone Marrow Endothelial Cells and Progenitors"

_ijms, 2024, doi:10.3390/ijms252212047_

Round 1
Reviewer 1 Report
Comments and Suggestions for Authors
Overall, it is very interesting manuscript that focus on endothelial cells as a compartment of the bone marrow microenvironment of multiple myeloma. In this manuscript, the authors used already published dataset from the Cupedo lab and performed additional analysis with a particular focus on endothelial cells. They describe a protocol to culture the endothelial cells in vitro. Although the model does not fully recapitulate the findings from scRNA seq, it still can be improved, which is discussed by the authors in the Discussion section.
I have several points that should be checked or discussed:
To merge the scRNA seq datasets, was any harmonization algorithm applied?
From the scRNA seq data, the authors identified more of the incoming VEGF interactions with endothelial cells and more of the outgoing (efferent) interaction based on PDGF interactions. Could any of these be confirmed to be present in the primary EC derived from CD138 negative fraction from MM patients cultured in vitro (eg by qPCR or flow cytometry staining for the receptors), to validate the findings from scRNA seq data? The cells may be changed by the in vitro culture, but these experiments may indicate if ECs per se are changed in the BM of MM patients and if the modulations, observed by different interactions with neighbouring cells, are stable.
Line 388, methods section, please include more information about MM patients, that were used in the study. Were they newly diagnosed or relapsed? What was the success rate of ECs culture from the BM aspirates of MM patients?
Minor:
Line 60: Please change Fig. A1 to Fig. 1A
Line 92. At the end of line, a sentence begins: A complementary… but it seems not to have any ending.
Line 137: please adjust reference to Figures, A2 and A3 do not make sense, the same applies for lines 155-159, lines 283-284.
Figure 5 and 6, indicate how many biological replicates were used in these assays and if the data present mean or median, the error bars in Figure 5B represent SD or SEM?
Please include in the results section text references to Figure 6C and 6D.
Author Response
We appreciate the reviewer’s constructive comments, which have provided valuable insights for strengthening our manuscript. We have carefully considered each point and revised the manuscript accordingly to address the reviewer’s concerns.
Comments 1: Overall, it is very interesting manuscript that focus on endothelial cells as a compartment of the bone marrow microenvironment of multiple myeloma. In this manuscript, the authors used already published dataset from the Cupedo lab and performed additional analysis with a particular focus on endothelial cells. They describe a protocol to culture the endothelial cells in vitro. Although the model does not fully recapitulate the findings from scRNA seq, it still can be improved, which is discussed by the authors in the Discussion section.
Response 1: We thank the reviewer for their positive assessment of our manuscript and for recognizing the novelty of our work.
Comments 2: I have several points that should be checked or discussed:
To merge the scRNA seq datasets, was any harmonization algorithm applied?
Response 2: No, we did not apply a harmonization algorithm when merging the scRNA-seq datasets. Our analysis focused on comparing cell populations and gene expression patterns as they were originally captured in the datasets, allowing us to observe any inherent differences without introducing additional computational adjustments.
Comments 3: From the scRNA seq data, the authors identified more of the incoming VEGF interactions with endothelial cells and more of the outgoing (efferent) interaction based on PDGF interactions. Could any of these be confirmed to be present in the primary EC derived from CD138 negative fraction from MM patients cultured in vitro (eg by qPCR or flow cytometry staining for the receptors), to validate the findings from scRNA seq data? The cells may be changed by the in vitro culture, but these experiments may indicate if ECs per se are changed in the BM of MM patients and if the modulations, observed by different interactions with neighbouring cells, are stable.
Response 3: We conducted this validation experiment to confirm the scRNA-seq findings. Using primary cells derived from the CD138-negative fraction of MM BM, we assessed the expression of PDGFA by qPCR. Despite the potential alterations due to in vitro culture conditions, our results demonstrated that the MM EOCs and ECs retain the expression of this gene, which is in agreement with our predicted ligand-receptor interactions. The results are presented in Figure S4. We have updated the Results section accordingly.
Comments 4: Line 388, methods section, please include more information about MM patients, that were used in the study. Were they newly diagnosed or relapsed? What was the success rate of ECs culture from the BM aspirates of MM patients?
Response 4: We obtained anonymized MM samples from patients who were newly diagnosed. The success rate of generating EC cultures was 90%. We have updated the Methods section in order to contain this information.
Comments 5: Minor:
Line 60: Please change Fig. A1 to Fig. 1A
Response 5: Thank you for catching this mistake. We meant to reference Figure S1, and this has now been updated in the main text.
Comments 6: Line 92. At the end of line, a sentence begins: A complementary… but it seems not to have any ending.
Response 6: This mistake in Word document editing has now been corrected.
Comments 7: Line 137: please adjust reference to Figures, A2 and A3 do not make sense, the same applies for lines 155-159, lines 283-284.
Response 7: We apologize for this editing error. We intended to refer to Figure S2 and Figure S3. Due to change in naming convention between manuscript versions the names stopped matching. We have corrected this mistake.
Comments 8: Figure 5 and 6, indicate how many biological replicates were used in these assays and if the data present mean or median, the error bars in Figure 5B represent SD or SEM?
Response 8: We thank the reviewer for this comment. These important details have now been included in the legends of both Figure 5 and 6.
Comments 9: Please include in the results section text references to Figure 6C and 6D.
Response 9: We have now included references to both Fig. 6C and D.

Reviewer 2 Report
Comments and Suggestions for Authors
The manuscript entitled: “Characterization and Experimental Use of Multiple Myeloma Bone Marrow Endothelial Cells and Progenitors” (ID: ijms-3270178) by Garbicz et al. performs the first unbiased analysis of transcriptomic changes in MM -derived ECs compared to normal bone marrow ECs.
Albeit the manuscript is well written and of interest, minor comments should be addressed to further improve the manuscript.
Comments:
1. The discussion section should be more balanced according to strengths and weaknesses of the analyses.
2. Discussion section: the authors should more highlight how this vitro model provides a valuable tool for specific studies and how future studies should be designed.
3. Figure 1 B, Figure 2 D, E and Figure 3 must be enlarged to make it more readable.
Author Response
Response to Reviewer #2
We appreciate the reviewer’s constructive comments, which have provided valuable insights for strengthening our manuscript. We have carefully considered each point and revised the manuscript accordingly to address the reviewer’s concerns.
Comments 1: The manuscript entitled: “Characterization and Experimental Use of Multiple Myeloma Bone Marrow Endothelial Cells and Progenitors” (ID: ijms-3270178) by Garbicz et al. performs the first unbiased analysis of transcriptomic changes in MM -derived ECs compared to normal bone marrow ECs.
Albeit the manuscript is well written and of interest, minor comments should be addressed to further improve the manuscript.
Comments:
- The discussion section should be more balanced according to strengths and weaknesses of the analyses.
Response 1: We appreciate reviewer’s comment. We have rewritten parts of the Discussion in order to better reflect the balance between weaknesses and analyses.
Comments 2: 2. Discussion section: the authors should more highlight how this vitro model provides a valuable tool for specific studies and how future studies should be designed.
Response 2: This information is now exposed in a separate Discussion paragraph (lines 347-353).
Comments 3: 3. Figure 1 B, Figure 2 D, E and Figure 3 must be enlarged to make it more readable.
Response 3: We agree with reviewer’s comment. We have enlarged these figures in the new version of the manuscript.

Reviewer 3 Report
Comments and Suggestions for Authors
This study conducted gene expression and pathway analysis of ECs derived from MM and HD using previously published scRNA-seq datasets. It further established a EPCs and ECs culturing method directly from MM patient BM. Their function and role in promoting MM proliferation were confirmed by in-vitro assays. This is a well-written manuscript with extensive amounts of data. It also provides novel EPCs and ECs culturing methods in this field. It can be further improved by addressing the following concerns.
Line 58 and 59, CD45+CD38+ cells are shown twice in this sentence, please double-check the cell markers expression in those datasets.
To analyze relative gene expression levels from real-time PCR data, I recommend using 2-ΔΔCT method and normalizing the expression level of control group to 1. So, it would be clearer how much fold of increase or decrease in each group vs the control group.
Author Response
We appreciate the reviewer’s constructive comments, which have provided valuable insights for strengthening our manuscript. We have carefully considered each point and revised the manuscript accordingly to address the reviewer’s concerns.
Comments 1: This study conducted gene expression and pathway analysis of ECs derived from MM and HD using previously published scRNA-seq datasets. It further established a EPCs and ECs culturing method directly from MM patient BM. Their function and role in promoting MM proliferation were confirmed by in-vitro assays. This is a well-written manuscript with extensive amounts of data. It also provides novel EPCs and ECs culturing methods in this field. It can be further improved by addressing the following concerns.
Line 58 and 59, CD45+CD38+ cells are shown twice in this sentence, please double-check the cell markers expression in those datasets.
Response 1: We thank the reviewer for the comment. This is intentional and derived from the original scRNAseq datasets from the Cupedo lab. De Jong et al. published a dataset containing CD45+ CD38+ immune cells (mostly lacking PCs) as well as a separate dataset containing CD45+ CD38+ plasma cells. We have added this note to the Results text in order not to confuse the reader.
Comments 2: To analyze relative gene expression levels from real-time PCR data, I recommend using 2-ΔΔCT method and normalizing the expression level of control group to 1. So, it would be clearer how much fold of increase or decrease in each group vs the control group.
Response 2: We appreciate the suggestion to use the 2^-ΔΔCT method for analyzing relative gene expression and normalizing the control group to 1 for fold-change comparison. However, we chose the 2^-ΔCT approach because it allows us to directly compare expression levels between multiple groups without relying on a single reference or control sample as a calibrator. This method provides a clear, relative quantification of expression in each condition independently, which is useful for the current study’s comparisons. Moreover, using 2^-ΔCT avoids potential biases that could arise from assuming a fold-change of 1 in the control group when, in fact, variability exists within biological samples. To ensure transparency and ease of comparison, we also provide statistical analyses directly on the 2^-ΔCT values, which account for experimental variation without requiring conversion into a fold-change relative to the control. We believe this approach offers a more accurate reflection of the data’s variability across groups, especially given our study’s focus on multiple experimental conditions rather than a strict control-versus-treatment setup.
